

# Evidence of validity and reliability of Jumpo 2 and MyJump 2 for estimating vertical jump variables

Amilton Vieira[1], Gabriela L. Ribeiro[1], Victor Macedo[1], Valdinar de Araújo Rocha Junior[1], Roberto de Souza Baptista[2], Carlos Gonçalves[1], Rafael Cunha[1] and James Tufano[3]

[1] Strength and Conditioning Laboratory, Faculty of Physical Education, Universidade de Brasília, Brasília, Distrito Federal, Brazil
[2] Automation and Robotics Laboratory, Faculty of Technology, Universidade de Brasília, Brasília, Distrito Federal, Brazil
[3] Faculty of Physical Education and Sport, Charles University Prague, Prague, Czech Republic

## ABSTRACT

**Background**. We investigated the concurrent validity and test-retest reliability of the Jumpo 2 and MyJump 2 apps for estimating jump height, and the mean values of force, velocity, and power produced during countermovement (CMJ) and squat jumps (SJ).
**Methods**. Physically active university aged men ($n = 10$, $20 \pm 3$ years, $176 \pm 6$ cm, $68 \pm 9$ kg) jumped on a force plate (*i.e.*, criterion) while being recorded by a smartphone slow-motion camera. The videos were analyzed using Jumpo 2 and MyJump 2 using a Samsung Galaxy S7 powered by the Android system. Validity and reliability were determined by regression analysis, typical error of estimates and measurements, and intraclass correlation coefficients.
**Results**. Both apps provided a reliable estimate of jump height and the mean values of force, velocity, and power. Furthermore, estimates of jump height for CMJ and SJ and the mean force of the CMJ were valid. However, the apps presented impractical or poor validity correlations for velocity and power. Compared with criterion, the apps underestimated the velocity of the CMJ.
**Conclusions**. Therefore, Jumpo 2 and MyJump 2 both provide a valid measure of jump height, but the remaining variables provided by these apps must be viewed with caution since the validity of force depends on jump type, while velocity (and as consequence power) could not be well estimated from the apps.

## INTRODUCTION

The countermovement jump (CMJ) and squat jump (SJ) are not only exercises that are used in training, but they are also movements that are widely used to monitor neuromuscular performance (*van Hooren & Zolotarjova, 2017*). Due to the inherent differences between these movements, the CMJ usually results in greater performance (*i.e.*, ∼20% greater jump heights) mostly due the effects of the stretch-shortening cycle (*van Hooren & Zolotarjova, 2017*). As the SJ does not make use of the stretch-shortening cycle, the CMJ and SJ are

Corresponding author
Amilton Vieira, amiltonvieira@unb.br

often both assessed to provide a more holistic view of lower body explosive strength performance. By using a force plate to assess CMJ or SJ performance, several variables can be obtained that can provide useful information regarding the mechanical capabilities of the neuromuscular system. Although force plates are, in fact, becoming more accessible than they were in the past, the high cost associated with a force plate system still presents a major barrier to their use among the general public.

As an alternative, smartphone applications (app) have been developed that make use of mobile phone cameras instead of large expensive force plates. One of the most popular apps, MyJump 2 which operates on the iOS system, has been shown to provide valid and reliable measures of the height of CMJ and SJ (estimated from flight time) (*Balsalobre-Fernández, Glaister & Lockey, 2015*; *Coswig et al., 2019*; *Cruvinel-Cabral et al., 2018*; *Brooks, Benson & Bruce, 2018*). More recently, other studies found valid and reliable measures of both CMJ and SJ flight time when using the Jumpo app, which runs on an Android operating system, allowing for a far broader user base than apps on iOS (*Vieira et al., 2021*; *Azevedo et al., 2019*). As these apps directly measure flight time, it is logical that this data can be used to then estimate jump height. However, these apps take things a step further and claim to calculate the mean values of force, velocity, and power produced during the propulsive phase (*i.e.,* the ascendant aspect of the movement) of the vertical jump (*Samozino et al., 2008*).

These measures can be highly informative for those seeking to analyze the force-velocity capabilities of the lower limbs. However, a recent study has questioned the validity of Samozino's method of estimating force, velocity, and power (*Hicks, Drummond & Williams, 2021*). To briefly explain, *Hicks, Drummond & Williams (2021)* observed that Samozino's method overestimated force and underestimated velocity and power of the CMJ compared to the force plate. However, it is important to note that the proposed method by *Samozino et al. (2008)* was developed specifically for the SJ exercise, and since a difference exists between CMJ and SJ performance, a study applying Samozino's method in both CMJ and SJ is required to show whether the jump type might play a role in this discrepancy.

Lastly, the validated iOS MyJump app has recently released an Android version, and the Jumpo app released its second version on Android, but neither the validity nor the reliability of these apps have been determined. Android is the most popular operating system and, unlike iOS, runs on devices from different companies, which implies that these devices may have different hardware and software characteristics like camera resolution, file storage, *etc.* Therefore, the purpose of the presented study was to investigate the concurrent validity and test-retest reliability of the Jumpo 2 and MyJump 2, both running on an Android system, for estimating jump height and the mean values of force, velocity, and power produced during CMJ and SJ.

## MATERIALS & METHODS

### Participants

Ten physically active university-aged men ($20 \pm 3$ years, $176 \pm 6$ cm, $68 \pm 9$ kg) participated in this study since results from a pilot testing revealed that a high coefficient of validity (*i.e.,*
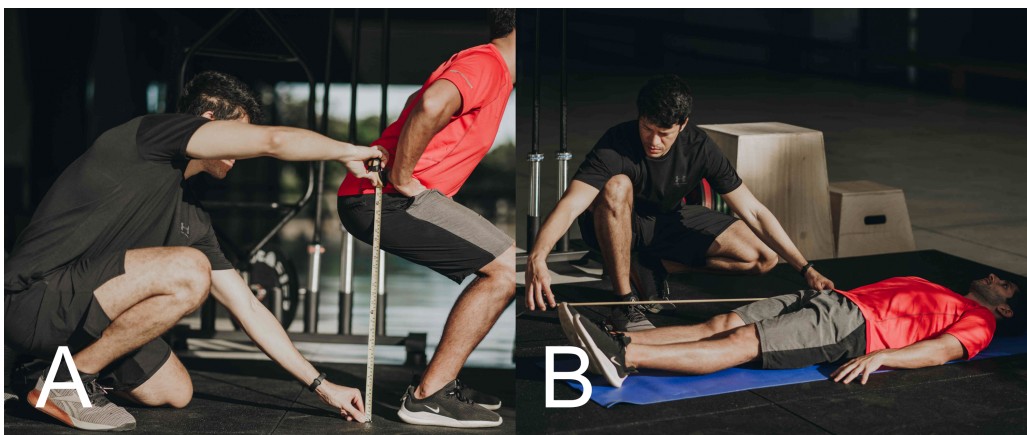

**Figure 1** **Anthropometric measurements for estimating the vertical displacement of the center of mass during the propulsion phase of vertical jumps.** (A) Measures were taken with the participant in a crouch position (90° knee angle set using a handheld goniometer) from the greater trochanter to the floor and (B) lying on a bench with lower limbs fully extended from the greater trochanter to tiptoe. The difference between these measures was used to estimate the center of mass vertical displacement.

$r > 0.9$) could be reached (*Hopkins & Rowlands, 2020*). They were classified as physically active by International Physical Activity Questionnaire (IPAQ) (*Craig et al., 2003*). They were free from any injury that could compromise jump performance. The participants were exercise science students and were well-acquainted with the CMJ and SJ movements. The UDF University Center Ethical Committee approved the study protocol (number 2.878.364) and each participant assigned a written informed consent.

## Study design
The participants visited the laboratory on three occasions (2 to 7 days apart). Firstly, anthropometric measurements were taken, such as the vertical displacement of the center of mass during the propulsion phase of the vertical jumps (SCoM; Fig. 1) (*Samozino et al., 2008*). In addition, the participants were familiarized with CMJ and SJ jumping technique and procedures, and subjects were considered to be familiarized when there was <1 cm difference in jump height between consecutive jumps of each jump type. In the following two visits, they completed ~10 min of a standardized warm-up protocol composed of traditional back squats (50% body mass) in a Smith Machine followed by five vertical jumps of progressive effort (*i.e.,* 20, 40, 60, 80 and 100% of the perceived maximum). In a quasi-randomized order, participants completed four repetitions of each jump type (countermovement jump [CMJ] and squat jump [SJ]), with all four jumps of one type followed by all four jumps of the other type. Jumps were performed on a 101 × 76 cm force plate (Accupower Portable Force Plate; AMTI, Watertown, MA, EUA) while being recorded by a slow-motion camera (240 fps) on a Samsung Galaxy S7 phone (model SM - G930F). Each jump was separated by 1 min of rest, and the same order of the jumps was kept for the re-test session for each participant.

## Jump performance

Participants performed each jump with hands akimbo and were instructed to jump as high as possible immediately following an auditory jump command ("three, two, one, jump"). They were asked to keep their legs straight during the flight phase of the jump and to land in the same place keeping the same posture as at takeoff. For CMJ, participants performed a downward movement followed by a vigorous hip extension, knee extension, and ankle plantar flexion. According to standard practice, they were free to determine their own countermovement depth for each jump. For SJ, they were required to achieve a squat position with ~90° of knee and hip flexion and hold this position for 2s before jumping. The force-time curve was inspected after each SJ, and if a counter movement occurred, the trial was repeated.

## Equipment and data analysis

The Jumpo 2 (https://play.google.com/store/apps/details?id=com.victormacedo10.JumPo2) and MyJump 2 (https://play.google.com/store/apps/details?id=com.my.jump) are apps available in the Google Play store (Google, Inc., Mountain View, CA, USA). Both apps make use of the mobile phone's camera and both calculate the flight time as an indicator of jump performance. The flight time was determined by selecting takeoff and landing frames moving the recorded video frame by frame using the interface of both apps. Based on flight time, the apps estimate jump height (*i.e.,* Center of mass (CoM) displacement) applying the constant acceleration equation (flight time$^2$ × 1.22625) (*Bosco et al., 1983*), and then based on jump height, body mass (m), gravitational acceleration (g), and SCoM the apps are able to estimate the mean values of force (Eq. (1)), velocity (Eq. (2)), and power (Eq. (3)) generated during the propulsive phase of the jumps (*Samozino et al., 2008*).

$$\bar{F} = mg\left(\frac{h}{\text{SCoM}} + 1\right) \quad (1)$$

$$\bar{v} = \sqrt{\frac{gh}{2}} \quad (2)$$

$$\bar{P} = \bar{F}\bar{v}. \quad (3)$$

To record the jumps, a tripod (30 cm high) with the smartphone attached on top was positioned 1.5 m away from the anterior aspect of the participant's feet. An independent observer, with no previous experience in video analysis (to maintain ecological validity of the apps being used by non-researchers in practical settings), was instructed to identify the last frame in which at least one foot was on the ground (*i.e.,* takeoff) and subsequently the first frame where at least one foot contacted the ground (*i.e.,* landing).

The force plate (*i.e.,* criterion method) was connected to a laptop equipped with a software (AccuPower 2.0.3; AccuPower, Dickinson, ND, USA) and recorded the ground reaction force (GRF) at 1,000 Hz. A posteriori, we analyzed the GRF with a custom-made MATLAB script (R2020a; MathWorks, Natick, MA, USA). Briefly, a low pass filter with cutoff frequency of 30 Hz and zero phase lag (Butterworth 4th order) was initially applied in the raw force signal (*Street et al., 2001*). After that, the CoM velocity was calculated using the trapezoid rule by numerically integrating time to acceleration (*Linthorne, 2001*). Force,

velocity, and power were obtained from the propulsive phase of the jump, and the jump height was estimated from flight time, as mentioned earlier.

## Statistical analyses

The average values (from four jumps) for each variable were analysed. Several analyses were conducted to determine the concurrent validity of both apps and the test-retest reliability of all devices. Comparisons between data from both apps and the force plate were made using one way ANOVA since all assumptions were met. When a significant difference was detected, the Tukey HSD *post-hoc* test was applied for multiple comparisons.

Regarding the concurrent validity, a regression analysis assessed the relationship between jump performance variables obtained from the force plate and Jumpo 2 app and again between the force plate and the MyJump 2 app (*Hopkins, 2004*; *Hopkins, 2015*). The validity correlations obtained from the regression analysis were interpreted as excellent (>0.995), very good (0.995–0.950), good (0.949–0.850), poor (0.849–0.700), very poor (0.699–0.450), and impractical (<0.450) (*Hopkins, 2015*). Raw and percentage (as a coefficient of variation –CV%) values of typical error of estimates were calculated and are displayed in Fig. 2. In addition, bias analysis (*i.e.,* uniformity of error) was performed by plotting residuals (predicted values minus "real" values obtained from force plate) against predicted values and then calculating a Pearson correlation between the residuals and predicted values.

Regarding the test-retest reliability, intraclass correlation coefficient (ICC) and typical error of measurement as a CV % were calculated. CV % were interpreted as good ($\leq$ 5%), moderate (5–10%), and poor (>10%) (*Hopkins, 2015*), while ICC were interpreted as poor (<0.500), moderate (0.500–0.750), good (0.749–0.900), and excellent (>0.900) (*Koo & Li, 2016*). Variables were assumed as "reliable" when the CV% was $\leq$ 10% and ICC was $\geq$ 0.75. IBM SPSS Statistics software (version 26; IBM Co., Chicago, IL, USA) was used for inferential statistics analyses, and a custom-made spreadsheet was used for validity and reliability (*Hopkins, 2015*).

# RESULTS

The regression analyses of jump variables obtained from the force plate and both apps are displayed in Fig. 2. Considering both CMJ and SJ, the validity correlations were "very good" for jump height for both apps against the force plate (Figs. 2A–2B). In contrast, we observed "poor" correlations for force (Figs. 2C–2D) and "very poor" correlations for velocity and power (Figs. 2E–2H). To note, CMJ demonstrated "good" to "very good" correlations for all the measured variables, except velocity, which presented a "poor" correlation. SJ demonstrated a "very good" correlation only for jump height, while force was "poor" and both velocity and power were "impractical" ($r < 0.410$). No evidence of heterocedascity (*i.e.,* non-uniformity of error) was observed for any variables since the Pearson correlations approached zero (slope < 0.01).

Comparisons between measures of the vertical jump obtained with the force plate and both apps are presented at Table 1. We found a "large" difference ($p = < 0.001$; $\eta^2 = 0.569$) between the mean velocity of the CMJ measured with the force plate compared with those

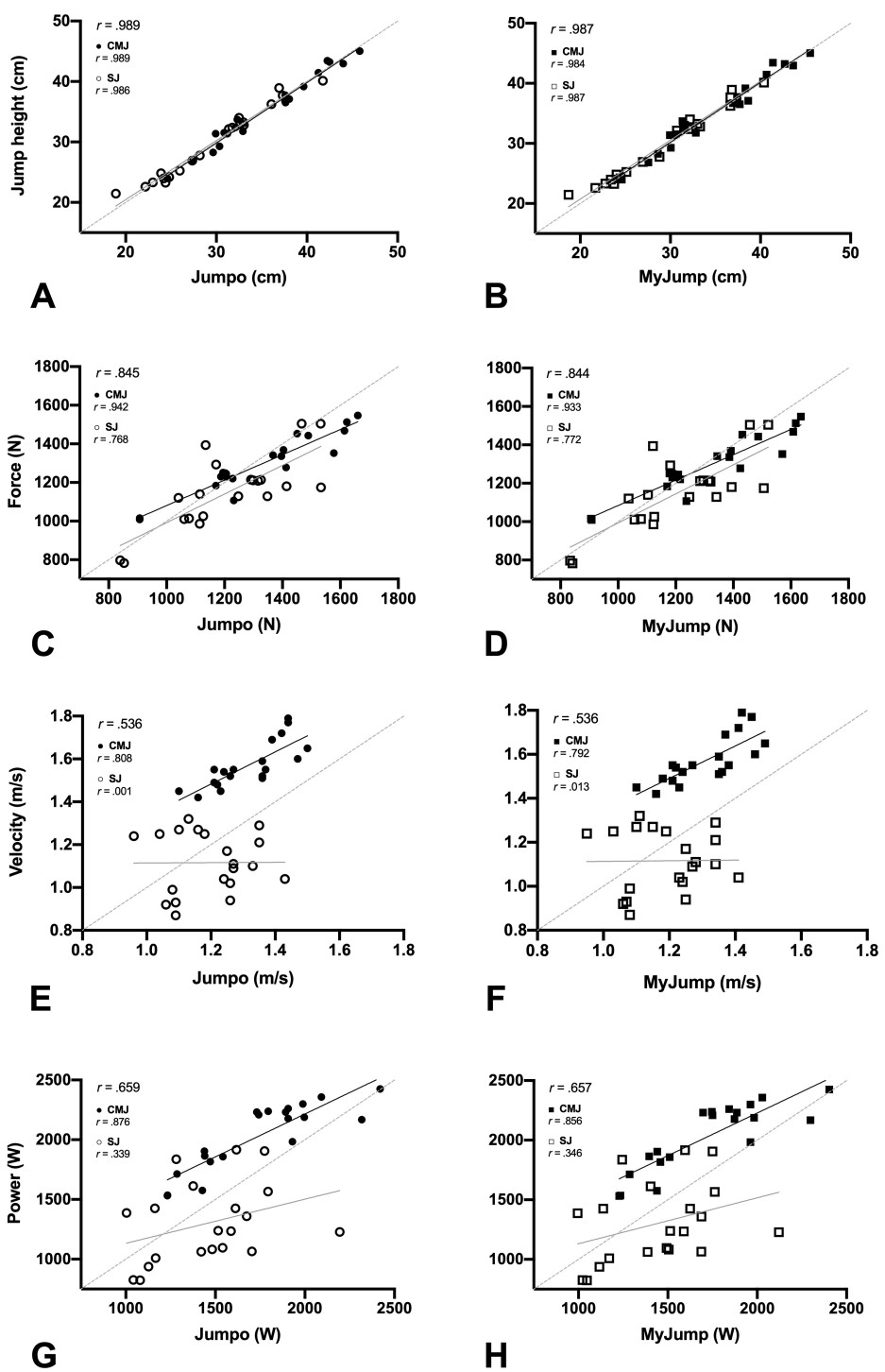

**Figure 2** **Regression analysis comparing Jumpo (A, C, E, G; *x*-axis) and MyJump (B, D, F, H; *x*-axis) against the force plate (criterion; *y*-axis).** The results on the top left of A–H display the validity correlations (r) combining both CMJ and SJ data.

measures that were estimated from both apps. We did not observe any other differences in the remaining variables.

The intra-rater reliability of the vertical jump measures obtained with the force plate and both apps is displayed at Fig. 3 and Table S1. Our results suggest that the majority of the measures show acceptable scores of reliability, and the measures obtained from CMJ were more likely to be more reliable than SJ. Note that the confidence intervals from SJ data more often extended past the CV and/or ICC limits compared to CMJ data.

## DISCUSSION

We investigated the concurrent validity and test-retest reliability of the Jumpo 2 and MyJump 2 apps for estimating jump height and mean values of force, velocity, and power generated during the propulsive phase of CMJ and SJ. No differences between Jumpo 2 and MyJump 2 were identified, but some large differences between the force plate and both apps were found (Fig. 2 and Table 1). Additionally, both apps provide a valid estimate of the jump height and mean propulsive force of the CMJ, whereas only jump height can be considered valid from the SJ (Fig. 2). Furthermore, the apps provide reliable estimates compared to the force plate (*i.e.,* criterion method), but the CMJ is more likely to have better scores of reliability than the SJ (Fig. 3).

The present findings corroborate with several previous studies demonstrating that the iOS version of the MyJump app (*Balsalobre-Fernández, Glaister & Lockey, 2015*; *Coswig et al., 2019*; *Cruvinel-Cabral et al., 2018*; *Brooks, Benson & Bruce, 2018*) and the original Jumpo app for Android (*Vieira et al., 2021*; *Azevedo et al., 2019*) provide valid measures of jump height estimated from flight time. Nevertheless, to our knowledge, this is the first study to demonstrate that mean force can also be accurately estimated from the CMJ using smartphone apps. Although this may be true, we found a large discrepancy between the mean propulsive velocity (and as consequence mean propulsive power) measured with the force plate against the estimated mean velocity with both the apps (Fig. 2 and Table 1). Impractical ($r^2 <0.001$) and poor ($r^2 = \sim0.64$) validity correlations were observed for both SJ and CMJ, respectively (Figs. 2E and 2F). Furthermore, the apps underestimated mean velocity during the CMJ (Table 1) by $\sim0.26$ m/s, which is quite substantial considering the normal mean propulsive velocity of jumping in this population. With this in mind, it is important to note that both apps estimate mean velocity (see Eq. (2)), which would ideally require a constant CoM acceleration and an accurate measure of jump height (*Samozino et al., 2008*). However, these assumptions might not be true because the CoM likely decelerates a few milliseconds before takeoff (probably due to the addition of the shank and foot segments to the accelerating mass (*McMahon et al., 2018*)) and the estimated jump height from flight time is prone to overestimate the peak of vertical displacement of the CoM (*Aragon-Vargas, 2000*). In support of this, a recent study (*Hicks, Drummond & Williams, 2021*) also found discrepancies in estimating CMJ mean velocity using Samozino's method. *Hicks, Drummond & Williams (2021)* observed that Samozino's method provided a poor validity correlation ($r^2 = 0.64$) and underestimated mean velocity by 0.26 m/s (16.4%) compared with the force plate. Note that we have found the same

Vieira et al. (2023), *PeerJ*, DOI 10.7717/peerj.14558

**Table 1** Comparisons of countermovement and squat jump propulsion phase variables between the force plate and the Jumpo 2 and MyJump 2 apps.

| Variables | Countermovement Jump | | | | | Squat Jump | | | | |
|---|---|---|---|---|---|---|---|---|---|---|
| | Force Plate | Jumpo | $TEE_{Jumpo}$ | MyJump | $TEE_{MyJump}$ | Force Plate | Jumpo | $TEE_{Jumpo}$ | MyJump | $TEE_{MyJump}$ |
| Jump height (cm) | $35.2 \pm 6.2$ | $35.4 \pm 6.1$ | 0.9 (2.8) | $35.0 \pm 6.1$ | 1.1 (3.4) | $29.7 \pm 6.0$ | $29.5 \pm 5.9$ | 1.0 (3.9) | $29.1 \pm 5.9$ | 1.0 (3.8) |
| Mean force (N) | $1{,}292 \pm 150$ | $1{,}322 \pm 215$ | 51.8 (4.1) | $1{,}315 \pm 212$ | 55.5 (4.4) | $1{,}152 \pm 190$ | $1{,}215 \pm 197$ | 125 (10.9) | $1{,}209 \pm 194$ | 124 (10.8) |
| Mean velocity (m s$^{-1}$) | $1.57 \pm 0.11$ | $1.31 \pm 0.11^{*}$ | 0.06 (4.0) | $1.31 \pm 0.11^{*}$ | 0.07 (4.2) | $1.12 \pm 0.14$ | $1.20 \pm 0.12$ | 0.14 (14.1) | $1.19 \pm 0.12$ | 0.14 (14.1) |
| Mean power (W) | $2{,}028 \pm 284$ | $1{,}740 \pm 346$ | 141 (6.9) | $1{,}721 \pm 340$ | 151 (7.5) | $1{,}302 \pm 335$ | $1{,}458 \pm 306$ | 324 (27.1) | $1{,}443 \pm 300$ | 323 (27.0) |

**Notes.**

Data are presented as mean $\pm$ SD.

*Less than force plate ($p < 0.001$). The typical error of estimates (TEE) are presented as raw units and percentage (%).

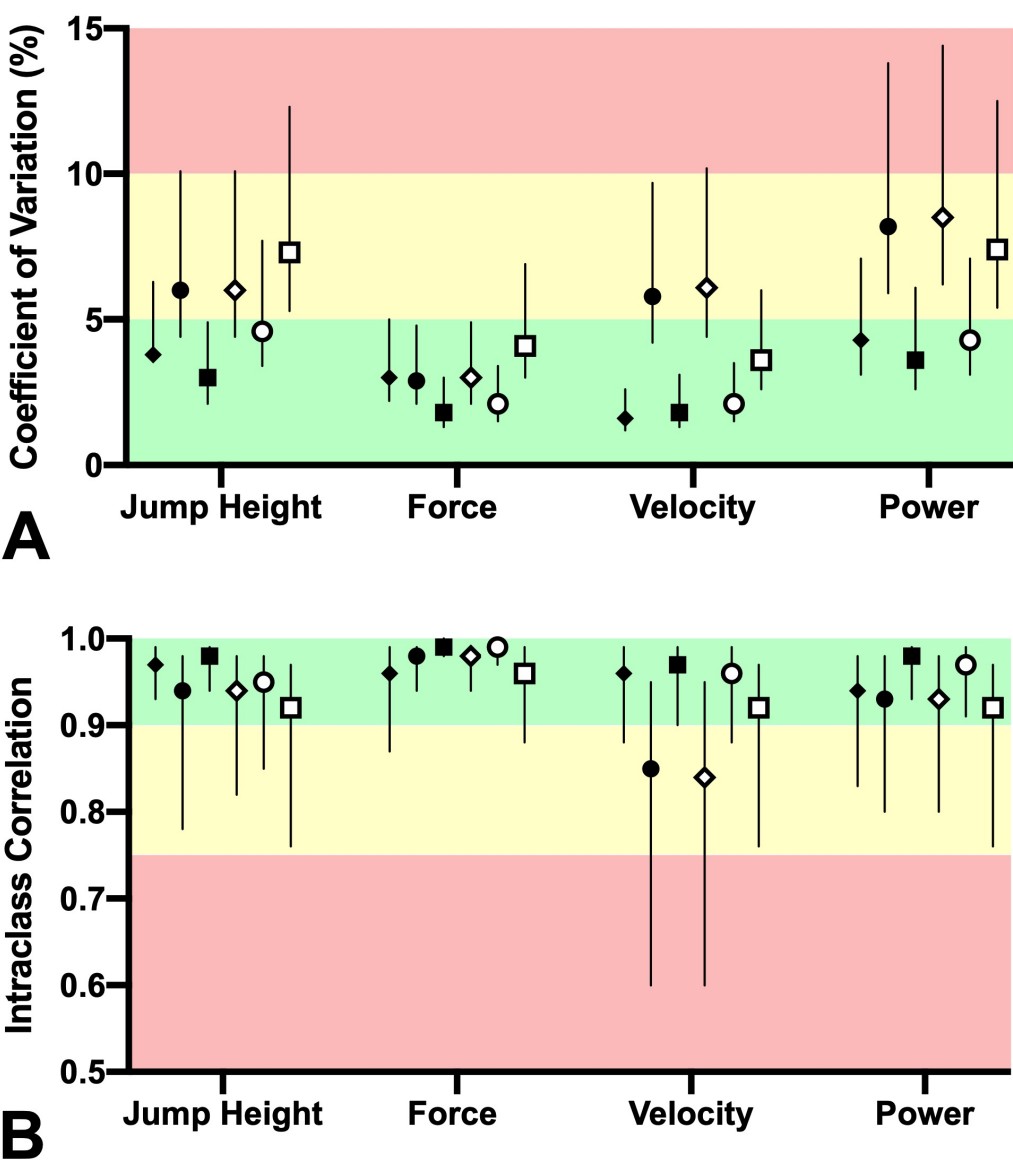

**Figure 3** **Intra-rater reliability of measures obtained during the countermovement (black symbols) and squat jumps (white symbols).** Force plate (◆ ◇), JumpO 2 (●○), and MyJump 2 (■ □) were used to measure these variables. Mean (symbol) with 90% CL (vertical lines) of (A) the coefficient of variations were interpreted as good (<5%, green), moderate (5% to 10%, yellow) and poor (>10%, red), while (B) intraclass correlation coefficient with 90% CL were interpreted as moderate (0.5 –0.75), good (0.75–0.90), and excellent (>0.90).

results for the CMJ, but even worse validity correlations for the SJ. In this regard, we could speculate that since CMJ performance is usually better than SJ performance, the SJ is more prone to be affected by the previously mentioned source of errors (lower signal to noise ratio).

Regardless of the jump type and the device used, there was an acceptable level of test-retest reliability for all the measured variables (ICC ≥ 0.75 and CV <10%, Fig. 3). It

was also noted that the CMJ (ICC $\geq$ 0.85 and CV $\leq$ 4.6%) demonstrated slightly better scores of reliability than the SJ (ICC $\geq$ 0.84 and CV $\leq$ 8.5%), which corroborates with previous studies demonstrating that the CMJ produces better reliability scores than the SJ (*Vieira et al., 2021*; *Gallardo-Fuentes et al., 2016*). Furthermore, both of the apps and the force plate demonstrated similar scores of reliability, which is an important finding considering the ecological validity of our methods and having an independent observer analyze the data without any previous experience in video analysis.

Considering all of the above, the present results have practical application for those seeking a simple approach to monitor and measure relevant performance variables with inexpensive mobile apps. Our findings revealed that the Jumpo 2 and MyJump 2 both provide reliable estimates of jump height and mean values of force, velocity, and power produced during the propulsive phase of both CMJ and SJ. However, the observed differences between force plate data and apps values for velocity and power of the CMJ as well as from all those variables except jump height from SJ prevent the users from confidently applying these measures in real life, for instance, to analyze the force-velocity capabilities during vertical jumps. It is worth mentioning that our sample size was small for those metrics beyond jump height (CMJ and SJ) and CMJ mean force due to their poor-to-impractical validity correlations. Furthermore, the present study included only physically active college students and further testing in other populations (*e.g.*, athletes, elderly, and sedentary individuals) is warranted before using these apps in those populations who may have better or worse jump mechanics and flight paths, as flight time is the main variable of interest when using these apps.

## CONCLUSIONS

Jumpo 2 and MyJump 2 mobile apps running on the Android system are reliable tools to monitor vertical jump performance variables, presenting scores of reliability similar to those observed with a force plate (*i.e.,* criterion equipment). Users should keep in mind that both apps provide valid measures of the height of CMJ and SJ (estimated from flight time) as well as the force produced during CMJ. However, a difference exists between velocity (and as consequence, power) when comparing force plate data against the estimated data from these apps.

### Funding

This work was supported by the Fundação de Apoio à Pesquisa do Distrito Federal (FAPDF) and Conselho Nacional de Desenvolvimento Científico e Tecnológico—CNPq, Processo: 438324/2018-8. The funders had no role in study design, data collection and analysis, decision to publish, or preparation of the manuscript.

### Grant Disclosures

The following grant information was disclosed by the authors:

Fundação de Apoio à Pesquisa do Distrito Federal (FAPDF)
Conselho Nacional de Desenvolvimento Científico e Tecnológico—CNPq, Processo: 438324/2018-8.

## Competing Interests

Amilton Vieira and Victor Macedo are designers of the Jumpo app. Data were obtained by an independent observer not related to the app development.

## Author Contributions

- Amilton Vieira conceived and designed the experiments, performed the experiments, analyzed the data, prepared figures and/or tables, authored or reviewed drafts of the article, and approved the final draft.
- Gabriela L. Ribeiro performed the experiments, analyzed the data, prepared figures and/or tables, authored or reviewed drafts of the article, and approved the final draft.
- Victor Macedo performed the experiments, analyzed the data, prepared figures and/or tables, authored or reviewed drafts of the article, Jumpo 2 app developer, and approved the final draft.
- Valdinar de Araújo Rocha Junior performed the experiments, analyzed the data, prepared figures and/or tables, authored or reviewed drafts of the article, and approved the final draft.
- Roberto de Souza Baptista analyzed the data, authored or reviewed drafts of the article, and approved the final draft.
- Carlos Gonçalves analyzed the data, authored or reviewed drafts of the article, and approved the final draft.
- Rafael Cunha analyzed the data, prepared figures and/or tables, authored or reviewed drafts of the article, and approved the final draft.
- James Tufano analyzed the data, prepared figures and/or tables, authored or reviewed drafts of the article, and approved the final draft.

## Human Ethics

The following information was supplied relating to ethical approvals (i.e., approving body and any reference numbers):

The Research Ethics Committee (Comitê de Ética em Pesquisa - CEP - Plataforma Brasil) granted ethical approval to carry out the study. (Protocol number: 2.878.364).

## Data Availability

The raw data are available in the Supplementary Files.

## Supplemental Information

Supplemental information for this article can be found online at http://dx.doi.org/10.7717/peerj.14558#supplemental-information.

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
