# Peer review of "Evidence of validity and reliability of Jumpo 2 and MyJump 2 for estimating vertical jump variables"

_PeerJ, doi:10.7717/peerj.14558_

## Round 0.1 · original submission · Minor Revisions

The variables extracted from recorded videos are, indeed, an estimation. Then, it is necessary to discuss the limitations of the study. Clarify how SCoM was estimated. Figure 1 is not explanatory. Clarify the numerical integration cited at lines 141-142.

Reviewer 1 ·

Basic reporting

1_ Force plates don't cost north of 10000 USD, stating the price makes the app worth a lot more than it is. By making that comparison you are making apps more appealing to use. The cost is around 5000 not 10000 for the Force Plate.

Experimental design

no comments

Validity of the findings

no comments

Additional comments

I enjoyed the paper and its simplicity, well written and explained.

Reviewer 2 ·

Basic reporting

The English language is clear.
The introduction was able to present the study problematic.
The paper structure is conforms to journal guidelines.
The objective is according with methods and results.

Experimental design

Original research within journal scope.
The research question is relevant.
The investigation present ethical standard.
The Methods section (procedures) is well described with sufficient detail to replicate the study.

Validity of the findings

The rationale and benefit to literature is clearly stated.
Statistic well done.
Conclusion well stated.

Additional comments

The aim of the present study was to investigate the concurrent validity and test-retest reliability of the Jumpo 2 and MyJump 2 for estimating jump height and the mean values of force, velocity, and power produced during CMJ and SJ. I would like to thank you for the opportunity to review this research. The present research report is quite novel and interesting, however, I think that a few small details need to be improved, which I will mention below:

Introduction:
The introduction is clear and nicely shows the main rationale of the paper. However, several studies have shown the My Jump validity. Thus, I would like to know why My Jump for Android could present different validity in comparison with My Jump for IOS? The authors can clarify this question in the introduction.

Lines 40-42: Is there a reference for this statement?

Methods:
Line 91: How many jumps consecutives?
Statistical Analysis: Why the authors did not perform the Bland-Altman analysis? Since this analysis provide bias and trend of underestimation and overestimation of the methods (i.e., Apps).

Results:
Lines 170-171: Please clarify that this statement is about both jumps (CMJ and SJ) analysis.
Lines 191-195: I would like to see the values of reliability. The authors could present these values in the text, or in a new table (values of day 1 and day 2 for jump height, force, velocity and power).

Discussion:
The discussion is well developed and connects to the current literature. I suggest the reference below, which aimed to validity the Jumpo App in fighters.
- Azevedo et al. 2019. Application for mobile devices is a valid alternative for vertical jump height measurement in fighters. Rev Andal Med Deporte. 2019;12(2): 83-87. https://doi.org/10.33155/10.33155/j.ramd.2019.01.007

Kind Regards.

·

Basic reporting

This study is clear, well-written, well-referenced with relevant background and the structure follows the PeerJ standards. The figures and table are relevant, clear and well described.

Experimental design

The research is within the PeerJ scope with an important relevant question. The methods are described allowing replication and performed with technical and ethical standard.

Validity of the findings

All data have been provided and statistically sound with important findings and conclusion are well stated linked to the supporting results.

Additional comments

General Comments
Good introduction leading to the problem clearly.
Methods section is clear with proper references.
Results are clear and to the point. One small suggestion for the authors consideration is to reorder the first sentence of each paragraph just to avoid the impression that the figures or tables are the results itself instead of the ones, which contains the data. For example “Regression analysis data from… are displayed at Figure 2.”, “Comparison between vertical jump measurements are presented at Table 1”, Intra-rater reliability is displayed at figure 3”.
Discussions were straightforward and well referenced.
Conclusions are proper to the findings leading to the application the study aimed.

Specific Comments
Line53 – Change “One of the more popular Apps…” to “One of the most popular Apps…”
Line 155 – I believe it is supposed to be Figure 2, instead of Figure 1.
Line 159 – intraclass correlation coefficient (ICC 3,1). What does 3,1 stand for? Might be a typo.
Results Table 1 – I suggest changing Force (N) to Mean Force (N).

·

Basic reporting

Overall, the article is well written and structured.
The references used are updated, but could be expanded.
Some adjustments to the tables and figures could be made to improve the presentation of results.

Experimental design

The objetive of study was to investigate the concurrent validity and test-retest reliability of the JumpO 2 and MyJump 2 for estimating jump height and the mean values of force, velocity, and power produced during CMJ and SJ. Although validation studies of these apps already exist in the literature, the authors justify that the second version of these apps for Android still needs to be tested for validity and reliability.

The study design is adequate, and the statistical analysis is correct.

I just have a few considerations/questions:
- The sample (n) is quite small to talk about validity... despite the authors reporting a high coefficient of validity, it is necessary to know to what extent this reduced sample is causing type I error.
- Four attempts were performed for each type of jump (CMJ and SJ), however, it is not clear whether the average of these values or the best value was used for analysis.
- The authors used a force platform as reference equipment, so it is assumed that the jump height will be calculated by the Impulse–momentum method. However, it was calculated by the flight time, which despite being an option, we cannot say that it is a reference, being more susceptible to errors than using the double integration of the ground reaction force.
- Bland-Altman plots would be welcome to observe individual errors and the BIAS, in addition to diagnosing a possible systematic or random error in the data. This could be done both in the analysis of validity and reliability.

Validity of the findings

The results are well presented.
I only suggest that the measures of agreement between the methods (typical error, for example) are not presented together with the consistency analyzes (correlations - Figure 2), but presented together with table 1 (or in a new table).

Conclusions of the study are punctual according to established objectives.

---

## Round 0.2 · accepted · Accept

The authors have responded properly to all comments and made all necessary changes to meet the reviewers' requirements. Please, consider the change to the title of the manuscript suggested by Reviewer #4.

Reviewer 2 ·

Basic reporting

The English language is clear.
The introduction was able to present the study problematic.
The paper structure is conforms to journal guidelines.
The objective is according with methods and results.

Experimental design

Original research within journal scope.
The research question is relevant.
The investigation present ethical standard.
The Methods section (procedures) is well described with sufficient detail to replicate the study.

Validity of the findings

The rationale and benefit to literature is clearly stated.
Statistic well done.
Conclusion well stated.

Additional comments

The authors answer all questions.
The manuscript is ready.
Congratulations.

·

Basic reporting

The text is clear, the authors answered all the questions and did the appropriate changes.

Experimental design

Methods clear anf replicable.

Validity of the findings

Interesting findings stricted connected to the results.

Additional comments

The authors made the changes and answered my doubts. I believe this is a good study necessary to the world of science and sports.

·

Basic reporting

The corrections and adjustments made by the authors improved the quality of the manuscript. In my analysis this revised version is adequate and well structured and can be accepted for publication.

I have just one suggedtion for the title, to insert the word "evidence" at the beggining: "Evidence of validity and reliability of Jumpo 2 and MyJump 2 for estimating vertical jump variables". Perhaps this can "alleviate" a bit the problem of making validity inferences with a very small sample.

Experimental design

Methods ok.

Validity of the findings

Results, discussion and conclusion - ok

Additional comments

No additional comments.